# Sensory and Biological Potential of Encapsulated Common Bean Protein Hydrolysates Incorporated in a Greek-Style Yogurt Matrix

**DOI:** 10.3390/polym14050854

**Published:** 2022-02-22

**Authors:** Samantha Free-Manjarrez, Luis Mojica, Hugo Espinosa-Andrews, Norma Morales-Hernández

**Affiliations:** Tecnología Alimentaria, Centro de Investigación y Asistencia en Tecnología y Diseño del Estado de Jalisco, A. C., CIATEJ, Unidad Zapopan, Camino Arenero 1227, El Bajío, Zapopan 45019, Mexico; jfree@ciatej.mx (S.F.-M.); lmojica@ciatej.mx (L.M.); hespinosa@ciatej.mx (H.E.-A.)

**Keywords:** gel, common bean protein hydrolysates, Greek-style yogurt, complex coacervation, functional food

## Abstract

The work aimed to develop a gel as a protective barrier of common bean protein hydrolysates to be incorporated into a Greek-style yogurt and evaluate the sensory perception and biological potential. The gel was formed by complex coacervation and induced heat at a pH 3.5 and 3:1 biopolymer ratio (whey protein and gum arabic). The gel presented a 39.33% yield, low syneresis (0.37%), and a gel strength of 100 g*_f_*. The rheological properties showed an elastic behavior (G′ > G″). The gel with the most stable characteristics favored the incorporation of 2.3 g of hydrolysates to be added into the Greek-style yogurt. Nutritionally, the Greek-style yogurt with the encapsulated hydrolysates presented 9.96% protein, 2.27% fat, and 1.76% carbohydrate. Syneresis (4.64%), titratable acidity (1.39%), and viscoelastic behavior presented similar characteristics to the Greek-style control yogurt. The bitterness and astringency in yogurt with encapsulated hydrolysates decreased 44% and 52%, respectively, compared to the yogurt control with the unencapsulated hydrolysates. The Greek-style yogurt with the encapsulated hydrolysates showed the ability to inhibit enzymes related to carbohydrate metabolism (α-amylase (92.47%) and dipeptidyl peptidase-4 (75.24%) after simulated gastrointestinal digestion). The use of gels could be an alternative to transporting, delivering, and masking off-flavors of common bean protein hydrolysates in food matrices to decrease glucose absorption for type 2 diabetes patients.

## 1. Introduction

Functional foods have attracted the interest of the food industry and consumers due to their essential role in preventing non-communicable diseases [1,2,3]. The growing demand can be explained by the rising cost of medical care, increased life expectancy, and increased consumer interest in obtaining health benefits [1]. There is a wide variety of bioactive compounds obtained from natural sources, such as pulses, plants, fruits, or vegetables, that show beneficial effects on human health and are used in the formulation of functional foods [4,5]. Common beans have been considered a functional food due to their content of polyphenols, resistant starch, oligosaccharides, saponins, phytosterols, proteins, and bioactive peptides [6,7]. Several authors have reported that common bean protein hydrolysates could play an essential role in controlling or preventing type 2 diabetes mellitus through in vitro assays [8,9,10] as in vivo [11]. Despite the health benefits, as the protein hydrolyzes, it provides a bitter or astringent taste in foods [12,13]. Furthermore, these compounds could react and interact with other components during food processing and storage, changing their composition and potential health benefits. Therefore, encapsulation is an important technology for the stability of bioactive peptides in the development of functional foods. In addition to keeping their functionality stable, it can protect by reducing unpleasant notes such as bitterness and astringency when consumed [2,13,14,15,16]. Encapsulation consists of coating solids, liquids, or gases with different materials to release the bioactive compounds at different intervals, limiting unwanted reactions and improving the stability, bioavailability, and organoleptic properties of these compounds [13,17,18]. The choice of the most suitable encapsulation method depends on the type of core material and the final product characteristics where the encapsulated will be applied [19]. An encapsulation alternative is a gel; its properties can be manipulated to develop a food grade system for encapsulating bioactive ingredients, improving their stability and delivery. A gel is an intermediate between a solid and liquid with flow and elastic characteristics. It also contains water in its network of cross-linked biopolymer molecules [20,21]. Moreover, the gels can act as texture modifiers and fat or starch replacers to reduce calories in foods [22]. Complex coacervation is a spontaneous phase separation of two oppositely charged biopolymers mixed in specific concentration, pH, temperature, and ionic strength conditions [23]. The complex coacervates with encapsulated bioactives, and can be subsequently embedded in a gel by heat treatment to improve the stability of the enclosed bioactives [24]. Recent research demonstrated that the encapsulation of peptides is an alternative to stabilize and preserve the biological potential of bioactive compounds and improve food acceptance [25,26,27,28]. A mussel protein hydrolysate was developed using spray drying with maltodextrin and modified starch in instant noodles. The encapsulation masks the bitterness attributed to the hydrophobic amino acids released in the protein hydrolysis [29]. Casein hydrolysates were encapsulated by complex coacervation using a soy protein isolate and pectin; the bitter taste and stability were improved by encapsulation [30].

A healthy diet can prevent or control type 2 diabetes, improving blood glucose and lipid control [31]. The American Diabetes Association recommends eating low-fat, low-sugar, and low-salt foods, including Greek-style yogurt [32]. Greek-style yogurt is currently the fastest-growing product in the dairy industry and is obtained after draining whey. Its consumption is directly related to health benefits associated with its higher nutritional value, especially its high protein content [33,34,35].

Therefore, the aims of this work were: (1) to generate a gel as a protective barrier for common bean protein hydrolysates from a whey protein concentrate and gum arabic by complex coacervation; (2) to incorporate the gel with common bean protein hydrolysates into a Greek-style-yogurt; and (3) evaluate the sensory perception (bitterness and astringency) and biological potential of Greek-style yogurt with common bean protein hydrolysates.

## 2. Materials and Methods

### 2.1. Materials

Raw black bean (*Phaseolus vulgaris* L.) San Luis cultivar (2017) from Sombrerete, Zacatecas, Mexico (23°37′53.9″ N 103°38′29.5″ W). The dry beans were stored at 4 °C until use. Pepsin from porcine gastric mucosa (EC 232-629-3), pancreatin from porcine pancreas (EC 232-468-9), α-Amylase from porcine pancreas (EC 232-565-6), quinine monohydrochloride dihydrate (90%), and tannic acid were purchased from Sigma-Aldrich (St. Louis, MO, USA). DPP-IV Glo^®^ protease assay was purchased from Promega (Madison, WI, USA). Gum arabic (GA; *Acacia senegal*) was purchased from Reasol (Mexico City, Mexico) (2.3% protein, 3.7% ash content, 0.1% fat, and 93.8% carbohydrates content). Whey protein concentrate (WPC) was purchased from Hilmar (Vilher, Jalisco, Mexico) (84.1% protein, 5.1% ash content, 0.7% fat, and 10.0% carbohydrates content). Canola oil was purchased from a local market (Jalisco, Mexico). Culture YO-PROX 753 was purchased from Bioprox (Mexico City, Mexico). All other reagents were analytical grade.

### 2.2. Gel (W-GEL) Preparation

Biopolymers (WPC and GA) were dispersed in deionized water, and the solutions were stored for the night at 4 °C to ensure complete hydration. Both biopolymer solutions were adjusted at pH 3.5. The coarse emulsion was obtained, mixing GA with canola oil (ratio 2:1) using a high-shear disperser at 9,500 rpm for 5 min (Ultra-Turrax T25, IKA Works, Inc., Wilmington, NC, USA). The emulsion size was reduced (500–600 nm) using an ultrasonic processor (Model CL-18, serial no. 2012030312), applying 50% of amplitude for 15 min. A portion of the prepared emulsion was mixed with WPC at two concentrations (5% and 7.5%) of total solids and two ratios of WPC:GA (5:1 and 3:1) to form the complex coacervates. The four mixtures were left to rest at 50 °C for 2 h. Then, the samples were centrifuged at 1000× *g* for 15 min. The supernatants were removed, and the sediments were heated at 85 °C for 30 min and cooled at 4 °C [36]. Four different W-GELs were formed and refrigerated at 4 °C for 24 h and subsequently characterized.

### 2.3. W-GEL Physicochemical Characterization

#### 2.3.1. Yield (%)

The yield of the W-GEL was determined using the following equation of Rodríguez-Rodríguez et al. [37]:Yield (%) = ((m_o_-m_i_)/m_o_) × 100(1)
where m_o_ is the initial mass used to produce the W-GEL, and m_i_ is the dried W-GEL mass.

#### 2.3.2. Gel Strength (Firmness)

The W-GEL strength was performed using a Texturometer TA XT PLUS (Godalming, Surrey GU7 1YL, UK) by a method reported by Qin et al. [38] with some modifications. The W-GEL was compressed using a 25% deformation to obtain the gel strength using a cylindrical probe (P/0.5R). The EXPONENT Stable Micro Systems version 6.1.16.0 software was used to calculate gel strength.

#### 2.3.3. Syneresis

Syneresis was measured under a method reported by Banerjee and Bhattacharya [39]. The W-GEL was centrifuged in 50 mL graduated tubes at 890× *g* for 15 min. Then, the supernatant was separated and weighed. The syneresis was calculated in percentage with the following formula:Syneresis (%) = ((m_1_ − m_2_)/m_1_) × 100(2)
where m_1_ is the initial weight of the W-GEL and m_2_ is the final weight of the sample after removing the supernatant.

#### 2.3.4. Rheological Measurement

The measurements of the viscoelastic properties of the W-GEL were carried out according to a methodology reported by Li et al. [40], using stress control rheometer equipment (AR1000, TA Instrument, New Castle, DE, USA) with a cross-hatched geometry (40 mm diameter) and a gap of 1000 µm at 4 °C. An amplitude strain sweep (0.1–100%) was performed using 1 rad/s to find the viscoelastic linear region. Subsequently, a frequency sweep was performed at 0.5% strain, varying from 0.1 to 100 Hz. In addition, a dynamic stress sweep test was performed to find the yield stress of emulsion gels using oscillatory stress from 1000 to 4000 Pa and 0.5% strain at 4 °C.

### 2.4. Application of the W-GEL with BPH in a Greek-Style Yogurt

#### 2.4.1. Common Bean Protein Hydrolysates (BPH)

Common bean protein hydrolysates were obtained following the procedure proposed by Mojica et al. [11]. The enzymatic hydrolysis was carried out with alcalase/substrate 1:20 (*w*/*w*) and pH 7.0 at 50 °C for 2 h. Then, the BPH was lyophilized.

#### 2.4.2. Preparation of W-GEL with BPH (BPH-GEL)

The methodology was followed as in Section 2.2. A total of 2.3 g of BPH was initially dispersed in the canola oil and GA to continue the gel formation, and then was incorporated in a serving of Greek-style yogurt (125 g).

#### 2.4.3. Greek-Style Yogurt Preparation

Low-fat milk (10 g/L) was heated to 92 ± 1 °C for 3 min and subsequently cooled to 43 ± 1 °C in an ice bath. Greek-style yogurt was produced by inoculating the YO-PROX 753 culture and incubating at 43 ± 1 °C for 4 h. Then it was cooled to 4 ± 1 °C and kept there for 18 h. After refrigeration, Greek-style yogurt was drained for 18 h. BPH-GEL was incorporated into the yogurt and mixed gently for its total integration (BPH-GEL-yogurt) at the end. Greek-style yogurt without gel, BPH (control yogurt), and Greek-style yogurt with unencapsulated BPH (BPH-Ue-yogurt) were made for physicochemical and sensory comparison. The products obtained were packaged and stored at 6 ± 1 °C for 3 days before analysis [41].

### 2.5. Characterization of BPH-GEL-Yogurt and Control Yogurt 

#### 2.5.1. Syneresis

A portion (10 g) of yogurt was centrifuged at 1000× *g* for 10 min. The supernatant was separated and weighed, and the syneresis was expressed as a percentage by weight compared to the initial weight of the yogurt [41].

#### 2.5.2. Titratable Acidity

Yogurt variations were analyzed by titratable acidity and expressed in lactic acid percentage. A portion (18 g) of yogurt was diluted in distilled water (1:2). Phenolphthalein (0.5 mL) was added, and then yogurts were titrated with 0.1 N NaOH until a permanent pink color appeared for at least 30 s [42].

#### 2.5.3. Rheological Measurement 

Rheological properties were determined according to a method reported by Crispín-Isidro et al. [41]. A rheometer equipped (AR1000, TA Instrument, New Castle, DE, USA) with a cross-hatched geometry (40 mm diameter) with a gap of 1000 µm at 4 °C was used. Amplitude sweeps (0.01 to 100% deformation, 1 Hz) were carried out in both yogurts. Before measuring, yogurts were standing for 30 min to recover the structure and equilibrate the temperature at 4 °C.

#### 2.5.4. Confocal Microscopy of BPH-GEL and BPH-GEL-Yogurt

Microstructures were evaluated according to the method described by Schmitt et al. [43]. The WPC was labeled with fluorescein isothiocyanate (FITC). BPH was labeled with rhodamine isothiocyanate (RBITC). Once marked, the BPH-GEL was formed. A confocal microscopy DMRA2 (Leica Microsystems GmbH, Wetzlar, Germany) was used with two lasers at 458 and 532 nm excitation, with a detection window between 492 and 520 nm and 590 and 690 nm, respectively. Images were captured and managed using LAS X software (Leica Microsystems). Samples were observed with a 40× objective magnification and Leica immersion oil type F solution.

### 2.6. Sensory Evaluation of BPH-GEL-Yogurt and BPH-Ue-Yogurt

Sensory evaluation was carried out by ten trained judges (20–48 years old) who completed the training according to ISO 8586-2 [44]. An unstructured scale was used to determine the degree of bitterness and astringency of BPH-GEL-yogurt and BPH-Ue-yogurt, using quinine (0.015, 0.021 y 0.027 g/L) and tannic acid (0.2, 0.37 y 0.48 g/L) as references, respectively. The quantitative perception of bitterness and astringency attributes were determined for each sample.

### 2.7. Biological Potential of BPH-GEL-Yogurt and Control Yogurt

#### 2.7.1. Gastrointestinal Digestion In Vitro

The release of bioactive compounds was measured and simulated in vitro gastrointestinal conditions described by Sanguansri et al. [45]. Simulated gastric fluid (SGF) was prepared by dissolving pepsin (3.2 g/L) in sodium chloride (2.0 g/L) solution. The pH was adjusted to 2.0 with 2 N HCl. Simulated intestinal fluid (SIF) was prepared by dissolving pancreatin (10.0 g/L) in a monobasic potassium phosphate (6.8 g/L) solution. Subsequently, the pH was adjusted to 6.8 with 2 N NaOH.

BPH-GEL-yogurt and control yogurt were dispersed in SGF (10 mL) and incubated in a water bath for 2 h at 37 °C/100 rpm (Boekel Scientific, Feasterville, PA, USA). Subsequently, the pH was adjusted to 6.8 using 1M NaOH, and then SIF (10 mL) was added. The digestion was continued for three hours. Samples were taken at the end of the gastric and intestinal simulations. Samples were inactivated with hot water at 90 °C for 10 min and were finally dried (spray dryer) to carry out the corresponding tests.

#### 2.7.2. α-Amylase Inhibition Assay

An α-amylase inhibition assay was performed using a modified protocol by Kusano et al. [46]. The undigested starch was detected at 630 nm (blue, starch–iodine complex). Acarbose was used as a positive control. Percentage of inhibition was calculated using the formula:(3)$$\%In=\left(1-\left[A_{2}-A_{1}/A_{4}-A_{3}\right]\right)\times 100\%$$
where *A*_1_ is the absorbance of the incubated mixture containing hydrolysates sample, starch, and amylase, *A*_2_ is the absorbance of the incubated mixture of sample and starch, *A*_3_ is the absorbance of the incubated mixture starch and amylase, *A*_4_ is the absorbance of the incubated solution containing starch.

#### 2.7.3. DPP-IV Inhibition

A DPP-IV inhibition assay was performed using a method described by Hsieh-Lo et al. [47]. Luminescence was measured using a SpectraMax^®^ i3 Multi-Mode Detection Platform (Molecular Devices, Sunnyvale, CA, USA). The inhibition was calculated from the blank and enzyme control for each sample.

### 2.8. Statistical Analysis

All analyses were performed in triplicate. Data are expressed as the mean ± standard deviation. One-way analysis of variance (ANOVA) was performed using Statgraphics Centurion XVI 16.1.03 statistical software (StatPoint Technologies, Inc., Warrenton, VA, USA). Tukey test was performed to identify significant differences among treatments *p* < 0.05.

## 3. Results and Discussion

### 3.1. Physicochemical Properties of W-GEL

Four gels (W-GEL 1 to W-GEL 4) were formed by complex coacervation followed by induced heating, the combination of two WPC:GA ratios (5:1 and 3:1), and two final total solids concentrations (5 and 7.5%). No significant differences (*p* > 0.05) were observed among the yield of the W-GELs for treatments (Table 1). The firmness of a sample commonly used to describe the texture of food is associated with gel strength [48]. Treatments with a lower biopolymer concentration (5%) presented a higher gel strength, such as W-GEL 3, which presented the highest firmness with a value of 159 *g_f_* ± 8 in contrast to gels with higher biopolymer concentrations (7.5%), due to the relationship with the percentage of water retention. A high water loss produces firmer gel structures; on the contrary, less water release generates softer gels due to the amount of water entrapped in the gel network [49]. W-GEL textural properties can be associated with water retention capacity and the microstructure.

Syneresis is a natural phenomenon during which excess unbound water leaves the gel. This undesirable phenomenon can be reduced by selecting the appropriate biopolymer at a suitable final concentration, ratio, and pH [39]. The syneresis percentage ranged from 0.37 to 4.20%. W-GEL 1, W-GEL 2, and W-GEL 3 achieved the highest syneresis percentage. However, in W-GEL 4 significant differences (*p* < 0.05) were observed compared to the other treatments. Biopolymer content can influence the gel matrix’s syneresis, increasing the syneresis percentage values during storage [39,50]. This behavior was related to the ability of biopolymers, mainly proteins, to capture water, allowing the formation of a viscoelastic network with good water retention and increasing volume [39].

The linear viscoelastic region was observed at a strain of 1% for all samples, and a constant strain of 0.5% was chosen to carry out the frequency sweeps. All the W-GELs exhibited a predominantly solid viscoelastic behavior, since the elastic modulus (G’) was higher than the viscous modulus (G”) in the entire frequency range studied. Table 1 shows the rheological modulus G’ variation at low (0.1 Hz) and high (100 Hz) frequencies. The results showed that the elastic modulus remained constant at low and high frequencies, the typical behavior of a strong gel. The viscoelastic properties were significantly higher for W-GEL 1, W-GEL 3, and W-GEL 4 compared to W-GEL 2. This behavior has been related to the degree of ionization between both macromolecules, which move away from their stoichiometric relationship and cause a decrease in gel performance [51,52]. However, no significant differences (*p* > 0.05) were observed between W-GEL 1, W-GEL 3, and W-GEL 4; however, W-Gel 4 presented the lowest syneresis. For application in yogurt, W-GEL 4 was considered suitable for incorporating a dose of 2.3 g of BPH in 125 g of Greek-style yogurt; in addition, the proximal composition of W-GEL4 was 11.63% protein, 5.32% fat, 10.82% carbohydrates, and 0.24% minerals, its consistency favors its incorporation, and its low syneresis could have more stability during storage. Several applications of encapsulation by complex coacervation have favored the stability and perception of bioactive compounds in the development of functional foods [2,13]. Examples include orange essential oil [53] and terpenes in black pepper essential oil [54], in both cases improving their stability. The combination of biopolymers can be more efficient to encapsulate bioactive peptides or hydrolysates to mask bitter flavors and improve their stability and bioavailability [18]. Examples include gelatin–pectin to encapsulate lycopene [28] or chitosan–pectin and chitosan–xanthan gum to palm oil as wall materials for application in food systems in yogurt [55].

### 3.2. Application of the W-GEL with BPH in a Greek-Style Yogurt

#### 3.2.1. Characterization of BPH-GEL-Yogurt and Control Yogurt

The bromatological composition of the BPH-GEL-yogurt and the control yogurt are shown in Figure 1. The fat content was significantly higher in the BPH-Gel-yogurt than in the control yogurt (2.27 ± 0.02 vs. 1.97% ± 0.06 respectively; *p* < 0.05). This behavior was due to the addition of the BPH in the gel to retain more fat. The fat content in both products was lower compared with other studies that report a fat content above 6.4% (Costa et al. [56]) and 7.0% (Pinto et al. [57]) in natural Greek-style yogurt. Fat content is an important indicator of sensory attributes that impact the characteristics of this type of yogurt, such as consistency and the creaminess sensation [35,58,59].

Greek-style yogurt is mainly characterized by its high protein content, having double or triple the amount of protein compared to stirred yogurt [33,60,61]. The protein content in the BPH-GEL-yogurt significantly increased (*p* < 0.05) compared with the control yogurt. The protein increase was about 57% in the BPH-GEL-yogurt, due to the incorporation of BPH and WPC in the gel as an encapsulation system. A stirred yogurt protein is around 3.1% [62]. Considering that a Greek-style yogurt contains twice the protein, the results showed that the BPH-GEL-yogurt and control yogurt were within the standards. No significant differences (*p* > 0.05) were observed in carbohydrate content between the BPH-GEL-yogurt and control yogurt. The carbohydrate content in both yogurts is associated with the presence of lactose. The content was low because during whey drain lactose passes freely through the yogurt retention membrane, in addition to the fact that part of the lactose was metabolized during fermentation [59].

Significant differences (*p* < 0.05) were observed among the two yogurts in terms of syneresis (Figure 2). Syneresis was two-fold higher in the control yogurt (7.27% ± 0.13) than the BPH-GEL-yogurt (4.64% ± 0.88). This behavior was attributed to the BPH-GEL incorporation. The increase of proteins in the gel, attributable to WPC and BPH, played an essential role in the water-holding capacity, providing stability to the formed network [33,41].

Titratable acidity measures proteins and phosphates concentration in milk as a good hygienic-sanitary quality [63]. After the three days of processing, there was no significant difference between acidity values in the BPH-GEL-yogurt and control yogurt (1.40 ± 0.01 vs. 1.39 % ± 0.01; *p* = 0.29). A similar acidity behavior was reported with the addition of other ingredients such as inulin [64] or agave fructans [41] in low-fat yogurt.

Rheological measurements are considered the response at the macroscopic level of the properties at the microscopic level of food [65]. The rheological properties of yogurts are shown in Figure 3. All samples showed a predominantly elastic character over viscous behaviors (G′ > G″), indicating that non-relaxing protein bonds dominated over weak bonds that break and reform rapidly. The higher amount of non-relaxing protein bonds forms a much denser and stronger gel structure [59]. Protein-enriched yogurts (≥8%) have been reported to have a firmer texture [34]. This behavior is characteristic of a weak gel exhibiting thinning behavior, characterized by a linear viscoelastic region where G′ and G″ showed a constant value of a small percentage of deformation. This is followed by a linear non-viscoelastic region with a downward inflection of the moduli at higher strain values [41]. The rheological properties of yogurt were affected by the protein present in the matrix and the process of yogurt manufacture [34,41,66]. Both yogurts had a similar consistency. BPH-GEL-yogurt was not affected in texture and acidity characteristics with the addition of BPH-GEL. The structure and texture of the BPH-GEL-yogurt were favored by the increased protein content, showing good water retention in contrast to the control yogurt.

#### 3.2.2. Confocal Microscopy of BPH-GEL and BPH-GEL-Yogurt

The morphological characterization of the BPH-GEL and BPH-GEL-yogurt was performed by confocal microscopy. The identification of the BPH distributed in the gel is shown in Figure 4a,b. Figure 4a shows a cross-section of a fragment of the BPH-GEL, where the BPH (yellow color) is observed immersed in a W-GEL (blue color). The aggregates formed from the BPH-GEL are shown in Figure 4b. The microstructure of the gels was analyzed by confocal microscopy. This technique is used because it has a higher sharpness to stain the biopolymers and thus identify the spatial array structures of the biopolymers [67]. The presence of hemispherical particles (color blue) of sizes around 2–4 µm (Figure 4a) was observed. Furthermore, aggregated particles of different sizes (Figure 4b) were observed in the Greek-style yogurt (gray color), having aggregates smaller than 20 µm (Figure 4c), allowing a more homogeneous distribution in the product. This indicates that the encapsulation system guarantees the protection of these bioactive compounds when incorporated into food. Geremias-Andrade et al. [68] developed and characterized polysaccharide and protein gels with curcumin using confocal microscopy. Moreover, whey protein complexes and gum arabic were observed using confocal microscopy to evaluate the morphology and stability of the oil droplets [69].

### 3.3. Sensory Evaluation of BPH-GEL-Yogurt and BPH-Ue-Yogurt

One of the characteristics of protein hydrolysates and peptides is their bitter and astringent (dry mouth) perception [12,13], which makes them unpalatable, making it a challenge for their incorporation in food. These two descriptors were evaluated in BPH-Ue-yogurt and BPH-GEL-yogurt. In both, the addition of BPH was 2.3 g per serving (125 g). The bitter perception of BPH was mainly detected as opposed to astringency (Figure 5). In BPH-GEL-yogurt, both attributes’ perception was significantly lower than in BPH-Ue-yogurt (p < 0.05). The encapsulation process reduced 44% of the bitterness and 52% of the astringency of BPH in the BPH-GEL-yogurt. Some descriptors imparted by BPH were mushroom, liver, grass, umami, and butter in BPH-Ue-yogurt, in contrast to BPH-Gel-yogurt, in which acidic, salty, and fatty notes were more predominant. Grass and mushrooms were less perceived. The liver, grass, mushroom, and salty notes of the BPH are possibly attributed to their extraction process, due to an excess of salts. In addition, one of the characteristics of legumes is their earthy flavor—cooked and smoked potatoes—which reduce their acceptance [70]. Despite many nutritional characteristics, the use of bioactive peptides is limited due to their bitterness and astringency. Recent research demonstrated that encapsulation is an alternative to improve food acceptance [27]. The extract of the bitter gourd fruit is rich in antioxidants and antidiabetic compounds. Its encapsulation reduced its bitterness and improved the stability of its bioactive compounds to use in food formulations [71].

### 3.4. Biological Potential

Mojica and Gonzalez de Mejía [72] established the best processing conditions to generate BPH with in vitro antidiabetic potential (inhibition of α-amylase, α-glucosidase, and DPP-IV). Furthermore, Mojica et al. [11] evaluated the hypoglycemic capacity of the BPH in vivo, establishing the effective dose in rats to reduce 24.5% of the total area under the postprandial glucose curve (200 mg BPH/kg BW) and the equivalent dose in humans (2.3 g BPH in a 70 kg person). Thus, this dose was considered for incorporation into BPH-GEL to be added to a serving of Greek-style yogurt (125 g). The BPH-GEL-yogurt and control yogurt were subjected to an in vitro simulated gastrointestinal digestion to evaluate the inhibition of α-amylase and DPP-IV. The development of functional ingredients and their incorporation into food are an alternative to combat public health problems through food. Fermented dairy products are ideal for patients with type 2 diabetes [32]. Patients with glucose metabolism alterations could benefit from consuming functional foods that block α-amylase and DPP-IV. The inhibition of α-amylase may improve postprandial glucose in patients with type 2 diabetes [72]. Furthermore, DPP-IV inhibition may prevent incretin-like glucagon-like peptide-1 (GLP-1) and glucose-dependent insulinotropic peptide (GIP) degradation, potentiating insulin secretion and improving glucose homeostasis [73]. Several reports have been published related to foods as a source of bioactive compounds with antidiabetic potential, including oats [74], *Capparis spinosa L* [75], rice bran [76], and milk [77]. The in vitro simulated gastrointestinal analysis showed that in the stomach, no conclusive results were observed for α-amylase and DPP-IV inhibition in the yogurt samples (Table 2).

This behavior could be because the BPH were encapsulated, and their functions could be limited. The inhibition occurred from the intestinal level for both enzymes (Table 2). Significant differences (*p* < 0.05) were observed after 3 h of simulated digestion in the α-amylase inhibition between the BPH-GEL-yogurt (92.47% ± 0.58) and control yogurt (89.03% ± 0.97) at the intestinal level. Similar behavior was observed for DPP-IV inhibition. The highest inhibition was 75.24% ± 1.43 in the BPH-GEL-yogurt compared to 69.13% ± 0.74 in the control yogurt.

The increased inhibition of α-amylase and DPP-IV could be due to the bioactive compounds being released from the matrix that protected them at the intestinal level, in addition to the possibility of synergy between the hydrolysates present in milk and BPH. Recently, in a clinical study, Ramos-Lopez et al. [3] reported that 5 g of BPH could reduce postprandial glucose in healthy humans in an oral glucose tolerance test. In this sense, two servings of BPH-GEL-yogurt per day could be an alternative to enhance the anti-diabetes effect in consumers.

## 4. Conclusions

Due to the interest in the consumption of functional foods, a gel based on gum arabic and whey protein concentrate was developed by complex coacervation and induced heat. The gel showed characteristics of a strong gel and low syneresis, similar characteristics to Greek-style yogurt. Common bean protein hydrolysates were incorporated into the gel as a protection system, homogeneously distributed throughout the Greek-style yogurt. This product developed with the gel enriched with the common bean protein hydrolysates maintained the texture and acidity characteristics of the control yogurt but with a higher protein content. The gel served as a protective barrier for the common bean protein hydrolysates, and sensory perceptions of bitterness and astringency were reduced. The off-flavors reduction could favor the acceptance of the consumption of this type of product. An increase in the inhibition of enzymes related to the treatment of type 2 diabetes was observed. Two servings per day of Greek-style yogurt with 2.3 g of encapsulated common bean protein hydrolysates could be an option for consumers seeking health-promoting alternatives. With the development of these gels, it may be feasible to evaluate the incorporation of other bioactive compounds to develop functional foods with better consumer acceptance in the dairy, meat, or bakery industries with health benefits for the control or prevention of non-communicable diseases.

## 5. Patents

MX/a/2019/014603. “Proceso de obtención de sistema de enmascaramiento de sabores amargos, astringencia o similares para uso en alimentos”.

## Figures and Tables

**Figure 1 polymers-14-00854-f001:**
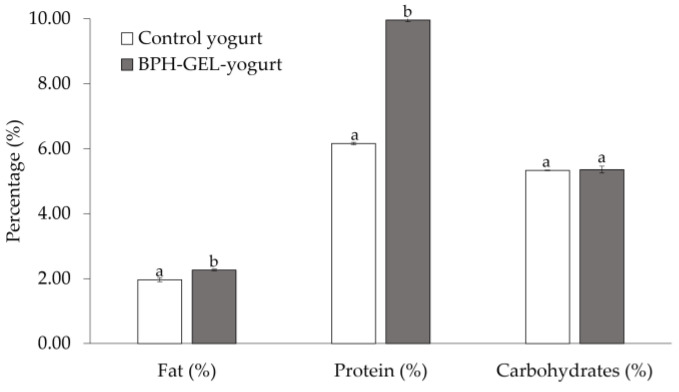
Bromatological composition of control yogurt and BPH-GEL-yogurt. BPH-GEL-yogurt: Greek-style yogurt with BPH encapsulated in a gel; control yogurt: Greek-style yogurt without gel and BPH. Values are the mean of two replicates ± standard deviation. Lowercase letters mean significantly different by parameter. (*p* < 0.05).

**Figure 2 polymers-14-00854-f002:**
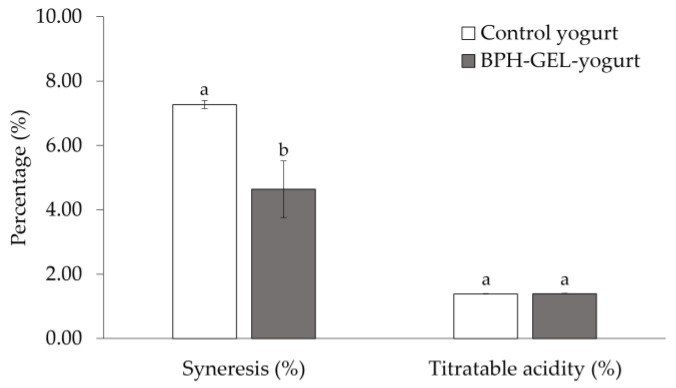
Physicochemical analysis of control yogurt and BPH-GEL-yogurt. BPH-GEL-yogurt: Greek-style yogurt with BPH encapsulated in a gel; control yogurt: Greek-style yogurt without gel and BPH. Values are the mean of three replicates ± standard deviation. Lowercase letters mean significantly different by parameter. (*p* < 0.05).

**Figure 3 polymers-14-00854-f003:**
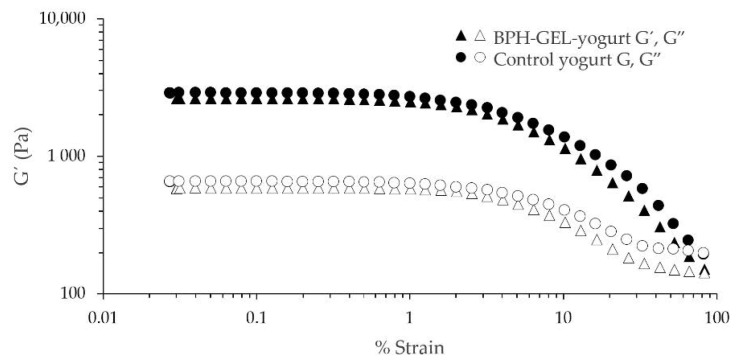
Rheological properties of control yogurt and BPH-GEL-yogurt. The evaluation was made in function of G′ and G″. BPH-GEL-yogurt: Greek-style yogurt with BPH encapsulated in a gel; control yogurt: Greek-style yogurt without gel and BPH.

**Figure 4 polymers-14-00854-f004:**
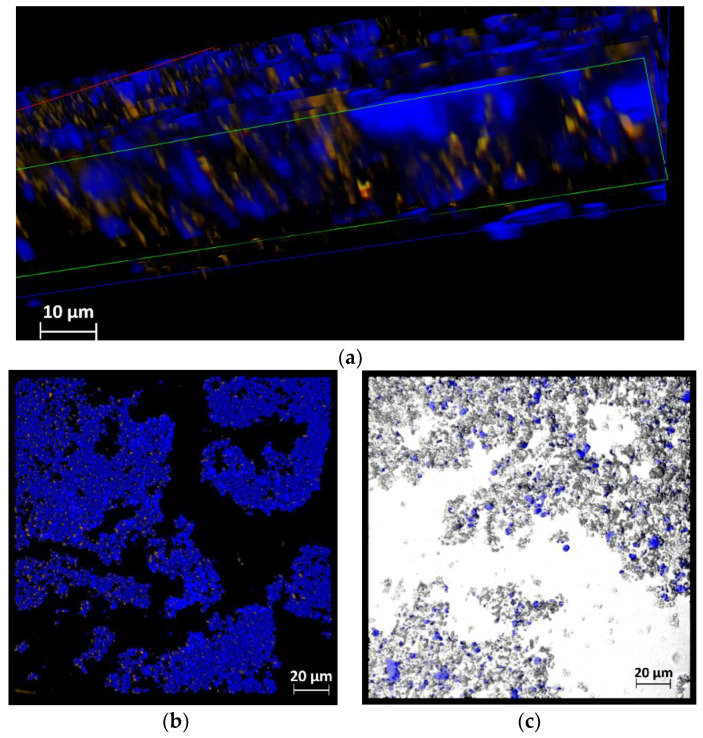
(**a**) Cross-section confocal microscopy of BPH-Gel, where BPH (yellow) is encapsulated in a gel (Blue). (**b**) Confocal microscopy of BPH-Gel (blue) encapsulating BPH (yellow). (**c**) Confocal microscopy of Greek-style yogurt (gray zone) with BPH-Gel, where BPH is encapsulated in a gel (blue).

**Figure 5 polymers-14-00854-f005:**
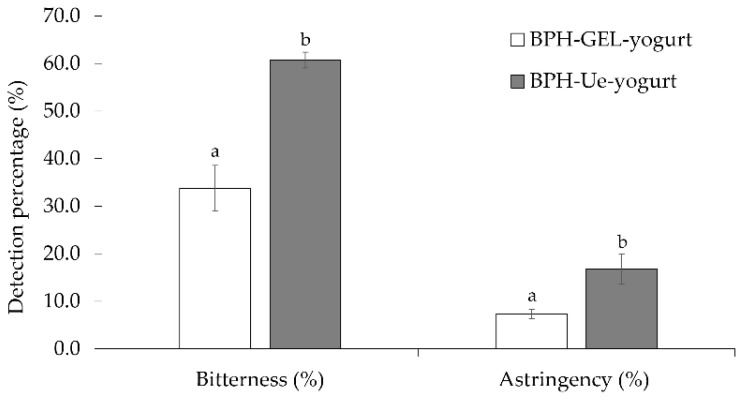
Bitterness and astringency of BPH-GEL-yogurt and BPH-Ue-yogurt. BPH-GEL-yogurt: Greek-style yogurt with BPH encapsulated in a gel; BPH-Ue-yogurt: Greek-style yogurt with unencapsulated BPH. Lowercase letters mean significantly different by the descriptor. (*p* < 0.05).

**Table 1 polymers-14-00854-t001:** Interrelation between yield, gel strength, syneresis, and rheology of W-GEL.

Tx.	Concentration	Ration.	Yield	Gel Strength	Syneresis	Rheology G′(Pa)	
	(%)	(WPC-GA)	(%)	(gf)	(%)	Ɯ (0.1)	Ɯ (100)
W-GEL 1	7.5	5:1	36.56 ± 1.17 ^a^	55 ± 2 ^a^	4.20 ± 0.05 ^a^	5722 ^a^	13,490 ^a^
W-GEL 2	7.5	3:1	35.78 ± 1.35 ^a^	96 ± 5 ^b^	2.71 ± 0.25 ^b^	3753 ^b^	9044 ^b^
W-GEL3	5.0	5:1	35.33 ± 1.89 ^a^	159 ± 8 ^c^	3.89 ± 0.62 ^a^	5473 ^a^	13,680 ^a^
W-GEL 4	5.0	3:1	39.33 ± 1.76 ^a^	100 ± 2 ^b^	0.37 ± 0.25 ^c^	5508 ^a^	13,730 ^a^

Tx: Treatment. W-GEL: WPC:GA gel without BPH. Values are the mean of three replicates ± standard deviation. Lower case letters mean significantly different by column. (*p* < 0.05).

**Table 2 polymers-14-00854-t002:** Biological potential of common bean protein hydrolysates. Inhibition of α-amylase and DPP-IV of BPH-GEL-yogurt and control yogurt.

Treatment	Sample	α-Amylase Inhibition	DPP-IV Inhibition
		(%)	(%)
BPH-GEL-yogurt	Initial	1.55 ± 1.61 ^aA^	ND
	Stomach	2.90 ± 0.18 ^aA^	ND
	Intestine	92.47 ± 0.58 ^bA^	75.24 ± 1.43 ^A^
Control yogurt	Initial	0.39 ± 4.37 ^aA^	ND
	Stomach	0.063 ± 3.5 ^aA^	ND
	Intestine	89.03 ± 0.97 ^bB^	69.13 ± 0.74 ^B^

Sample Concentration: 1 mg protein hydrolysate/mL. BPH-GEL-yogurt: Greek-style yogurt with BPH encapsulated in a gel; control yogurt: Greek-style yogurt without gel and BPH. Values are the mean of three replicates ± standard deviation. Lowercase letters (a, b) in the same treatment and capital letters (A, B) by type of samples means significantly different (*p* < 0.05).

## Data Availability

Data will be available upon request.

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
