# Peer review of "Sensory and Biological Potential of Encapsulated Common Bean Protein Hydrolysates Incorporated in a Greek-Style Yogurt Matrix"

_polymers, 2022, doi:10.3390/polym14050854_

Round 1

Reviewer 1 Report

This study aimed to Generate a gel as a protective barrier from whey protein and gum Arabic for common bean protein hydrolysates and to evaluate the sensory perception and biological potential of Greek-style yogurt. The research work topic is important and worth investigation and approval, however, there are many shortcomings that must be rectified. In general, important information is presented.

1. Please order the keyword as a function of their apparition in the abstract”.

  1. Please do not use abbreviations in the abstract
  2. This work has a variety of data that are not apparent by just reading the abstract. There appears to be some information, which can add to knowledge in this growing field.
  3. Pleases delete these keywords: “bitter; astringent; type 2 diabetes”
  4. Introduction: the originality is not clear! Please develop
  5. P8: please give more explication!
  6. In the discussion, the author would have benefited from a better understanding of the existing literature
  7. In some sentences, English appears not to be adequate.
  8. To use a space between the number and the unit, as 20 °C; and not to use a space between number and percentage, like 10%, for example.
  9. Conclusion: please include limitations and future research areas!
  10. Please include a list of abbreviation
  11. References must be revised.

Author Response

Thank you for the comments. We appreciate it.

  1. Please order the keyword as a function of their apparition in the abstract

Author response:

The order of the keywords was modified.

Keywords: Gel, common bean protein hydrolysates, Greek-style yogurt, complex coacervation, functional food.

  1. Please do not use abbreviations in the abstract

Author response:

All abbreviations have been removed from the abstract.

Some words were incorporated to complete the statement due to the elimination of abbreviations; they are easily visible with the track change function in the abstract.

Abstract: The work aimed to develop a gel as a protective barrier of common bean protein hydrolysates to be incorporated into a Greek-style yogurt and evaluate the sensory perception and biological potential. The gel was formed by complex coacervation and induced heat at pH 3.5 and 3:1 biopolymer ratio (whey protein and gum Arabic). The gel presented 39.33% yield, low syneresis (0.37%), and gel strength of 100 gf. Rheological properties showed elastic behavior (G’>G”). The gel with the most stable characteristics favored the incorporation of 2.3 g of hydrolysates to be added into Greek-style yogurt. Nutritionally, Greek-style yogurt with encapsulated hydrolysates presented 9.96% protein, 2.27% fat, and 1.76% carbohydrate. Syneresis (4.64%), titratable acidity (1.39%) and viscoelastic behavior presented similar characteristics to Greek-style control yogurt. The bitterness and astringency in yogurt with encapsulated hydrolysates decreased 44% and 52%, respectively, compared to yogurt control with unencapsulated hydrolysates. Greek-style yogurt with encapsulated hydrolysates showed the ability to inhibit enzymes related to carbohydrate metabolism α-amylase (92.47%) and dipeptidyl peptidase-4 (75.24%) after simulated gastrointestinal digestion. The use of gels could be an alternative to transporting, delivering, and masking off-flavors of common bean protein hydrolysates in food matrices to decrease glucose absorption, for type 2 diabetes patients.

  1. This work has a variety of data that are not apparent by just reading the abstract. There appears to be some information, which can add to knowledge in this growing field.

Author response:

In the abstract, the conditions for gel preparation were incorporated to carry out the complex coacervation process. Sentences were added to read. Line 12-13.

The gel was formed by complex coacervation and induced heat at pH 3.5 and 3:1 biopolymer ratio (whey protein and gum Arabic)

  1. Pleases delete these keywords: “bitter; astringent; type 2 diabetes”

Author response:

Done. Keywords: Gel, common bean protein hydrolysates, Greek-style yogurt, complex coacervation, functional food.

  1. Introduction: the originality is not clear! Please develop

Author response:

The statement was restructured, highlighting the importance of encapsulation during the development of functional foods. Lines 43-46.

Therefore, encapsulation is an important technology for the stability of bioactive peptides in the development of functional foods. In addition to keeping their functionality stable, it can protect by reducing unpleasant notes such as bitterness and astringency when consumed.

In the objectives, abbreviations were eliminated and the complex coacervation process to form the gel with hydrolysates was mentioned. Lines 75-79:

Therefore, the aims of this work were 1) Generate a gel as a protective barrier for common bean protein hydrolysates from whey protein concentrate and gum Arabic by complex coacervation, 2) Incorporate the gel with common bean protein hydrolysates into a Greek-style-yogurt, and 3) Evaluate the sensory perception (bitterness and astringency) and biological potential of Greek-style yogurt with common bean protein hydrolysates.

  1. P8: please give more explication!

Author response:

The rheological behavior of yogurt with hydrolysates is associated with protein composition. Lines 319-323.

Both yogurts had a similar consistency. BPH-GEL-yogurt was not affected in texture and acidity characteristics with the addition of BPH-GEL. The structure and texture of the BPH-GEL yogurt were favored by the increased protein content, showing good water retention in contrast to the control yogurt.

  1. In the discussion, the author would have benefited from a better understanding of the existing literature

Author response:

The proximal composition of the gel before application in yogurt was incorporated. Other relevant studies in the field of complex coacervation  were included. Lines 263-272.

The proximal composition of W-GEL4 was 11.63% protein, 5.32% fat, 10.82% carbohydrate and 0.24% minerals, its consistency favors its incorporation, and its low syneresis could have more stability during storage. Several applications of encapsulation by complex coacervation have favored the stability and perception of bioactive compounds in the development of functional foods [2,13]. Examples such as orange essential oil [75], terpenes in black pepper essential oil [76]. in both cases improving their stability. The combination of biopolymers can be more efficient to encapsulate bioactive peptides or hydrolysates to mask bitter flavors, improve their stability and bioavailability [18], such as gelatin – pectin to encapsulate lycopene [28] or chitosan - pectin, chitosan - xanthan gum to palm oil as wall materials for application in food systems in yogurt [77].

References added:

Rojas-Moreno, S.; Cardenas-Bailon, F.; Osorio-Revilla, G.; Gallardo-Velazquez, T.; Proal-Najera, J. Effects of complex coacervation-spray drying and conventional spray drying on the quality of microencapsulated orange essential oil. Journal of Food Measurement and Characterization 2018, 12, 650-660, doi: 10.1007/s11694-017-9678-z.

Bastos, L.P.H.; Vicente, J.; dos Santos, C.H.C.; de Garvalho, M.G.; Garcia-Rojas, E.E. Encapsulation of black pepper (Piper nigrum L.) essential oil with gelatin and sodium alginate by complex coacervation. Food hydrocolloids, 2020, 102, 105605, doi: 10.1016/j.foodhyd.2019.105605.

Rutz, J.K.; Borges, C.D.; Zambiazi, R. C.; Crizel-Cardozo, M.M.; Kuck, L.S.; Noreña, C.P. Microencapsulation of palm oil by complex coacervation for application in food systems. Food Chemistry, 2017, 220, 59-66, doi: 10.1016/j.foodchem.2016.09.194

  1. In some sentences, English appears not to be adequate.

Author response:

English style was revised and improved in some sections of the manuscript.

  1. To use a space between the number and the unit, as 20 °C; and not to use a space between number and percentage, like 10%, for example.

Author response:

Spaces between numbers and units and percentages were revised.

Line 83: 4 °C

Line 97: 4 °C

Line 103: 5% and 7.5%

Line 106: 85 °C and 4 °C

Line 107: 4 °C

Line 132: 4 °C

Line 136: 4 °C

Line 141: 50 °C

Line 147: 92 ± 1 °C

Line 148: 43 ± 1 °C

Line 149: 4 ± 1 °C

Line 154: 6 ± 1 °C

Line 169: 4 °C

Line 172: 4 °C

Line 180: 40 x

Line 198: 37 °C / 100 rpm

Line 201: 90 °C

Line 295: 4.64% ± 0.88

Line 404: 92.47% ± 0.58

  1. Conclusion: please include limitations and future research areas!

Author response:

Done. Sentences were added to read. Line 437 – 439.

With the development of these gels, it may be feasible to evaluate the incorporation of other bioactive compounds to develop functional foods in dairy, meat, or bakery foods with health benefits for the control or prevention of non-communicable diseases and with better consumer acceptance.

  1. Please include a list of abbreviation

Author response:

Done. The list of abbreviations is presented in Lines 444-461.

Abbreviation

GA: Gum Arabic

WPC: Whey protein concentrate

BPH: Common bean protein hydrolysates

W-GEL: Whey protein and gum Arabic Gel without common bean protein hydrolysates

BPH-GEL: Whey protein and gum Arabic Gel with encapsulated common bean protein hydrolysates

BPH-GEL-yogurt: Greek-style yogurt with encapsulated common bean protein hydrolysates in a Whey protein and gum Arabic gel.

Control yogurt: Greek-style yogurt without whey protein and gum Arabic gel and common bean protein hydrolysates

BPH-Ue-yogurt: Greek-style yogurt with unencapsulated common bean protein hydrolysates

DPP-IV: Dipeptidyl peptidase-4

FITC: fluorescein isothiocyanate

RBITC: rhodamine isothiocyanate

SGF: Simulated gastric fluid

SIF: Simulated intestinal fluid

  1. References must be revised.

Author response:

All references were revised to be in the journal format.

Reviewer 2 Report

Congrats! The article is very well designed, wrote and it presents significant data from the field of protein hydrolysates use and some possible industrial applications.

Author Response

Thank you very much for your comments